

# Characteristics of soil microbial phospholipid fatty acids in artificial, black-soil mountain degraded, and natural grasslands in the source region of the Three Rivers

Lele Xie,  Yushou Ma,  Yanlong Wang,  Yuan Ma,  Xiaoli Wang and Ying Liu

Qinghai Provincial Key Laboratory of Adaptive Management on Alpine Grassland, Xining, Qinghai Province, Xining, China

Qinghai University Academy of Animal Husbandry and Veterinary Sciences, Key Laboratory of the Alpine Grassland Ecology in the Three Rivers Region, Qinghai Province, Xining, China, China

## ABSTRACT

**Background**. The source region of the Three Rivers is a concentrated distribution area of alpine grassland. Due to intensified human interference and unsustainable land use, the vegetation and soil in these grasslands have undergone severe degradation, resulting in extensive areas of secondary bare land known as "black soil beach". A specific form of this degradation is referred to as "black-soil mountain". To address soil degradation in this region, the establishment of artificial grassland has been proposed. Recent research on grassland restoration has increasingly focused on belowground processes, particularly the role of soil microbial communities in soil recovery.

**Methods**. This study quantitatively analyzed vegetation characteristics, soil nutrients, microbial community structure, and influencing factors across three grassland types: artificial grassland (AG), black-soil mountain degraded grassland (BG), and natural grassland (NG). Standard laboratory analyses and the phospholipid fatty acid (PLFA) method were employed.

**Results**. The establishment of artificial grassland significantly increased aboveground biomass and the contents of soil organic carbon, total nitrogen, and total phosphorus, while notably reducing species richness. A total of 29 PLFA biomarkers were detected across the three grassland types, with AG showing significantly higher biomarker content than BG and NG. Key PLFA biomarkers included 16:1 w7c, 18:1 w7c, 15:0 iso, 15:0 anteiso, 16:00, and 18:1 w9c. Among microbial groups, bacteria were most abundant, followed by fungi, actinomycetes (act), and arbuscular mycorrhizal fungi (AMF). Compared to BG, AG exhibited significantly lower G+/G− and saturated-to-monounsaturated fatty acid (Sat:mono) ratios. Correlation analysis revealed that total PLFA, bacterial (B), fungal (F), G+, and G−contents were significantly or highly significantly positively correlated with soil organic carbon, total nitrogen, and water content ($P < 0.01$; $P < 0.001$). Redundancy analysis (RDA) showed that two principal components explained 76.96% and 13.74% of the variation in microbial community structure, with soil organic carbon and total nitrogen identified as the main driving factors.

Corresponding author
Yushou Ma, mayushou@sina.com

**Conclusion**. The establishment of artificial grassland is an effective strategy for restoring black-soil mountain degraded grassland. Monitoring microbial PLFA diversity and composition provides a reliable index for assessing soil environmental changes and nutrient dynamics. However, even after five years of restoration, the soil functionality of artificial grassland does not fully recover to the level of natural grassland.

## INTRODUCTION

The source area of the Three Rivers is the most important ecological function area of the Qinghai-Tibet Plateau (*Wang et al., 2022*). In recent years, with the influence of global warming and human factors, the grassland in this area has been continuously degraded, and some areas have been fragmented and islanded, degenerating into "black soil beach". The plant diversity and productivity will also change after the grassland is seriously degraded (*Li et al., 2022a*; *Wu et al., 2022*). This type is a typical extremely degraded alpine areas, which is characterized by small species, low forage yield, low vegetation coverage and upper soil loss (*Liu et al., 2018*; *Saruul et al., 2019*). Because its bare soil is "black", it is called "black soil beach" (*Ma et al., 2008*). When the black soil beach spreads to the hillside or the top of the mountain, a black-soil mountain (slope) is formed (*Li et al., 2014*). This has seriously affected the development of animal husbandry and the health of natural ecosystems in the source areas of rivers and rivers. Grassland degradation presents overwhelming challenges to biodiversity, ecosystem services, and the socioeconomic sustainability of dependent communities (*Li et al., 2021*; *Li et al., 2025*). It is urgent to solve these problems and implement effective protection and restoration measures, which cannot be ignored (*Bardgett et al., 2021*; *Wang et al., 2022*). In order to restore the ecological function and service function of grassland ecosystem, corresponding restoration measures are taken in different types of degraded grassland (*Ren, Lü & Fu, 2016*). To alleviate the degradation of alpine grassland ecosystem, effective management can be adopted, such as limiting grazing, removing poisonous weeds, replanting and rebuilding (*She et al., 2024*; *Yin et al., 2021*). The most effective and direct restoration method is to develop artificial grassland and semi-artificial grassland (*Dong et al., 2010*).

In recent years, in order to alleviate the pressure of natural grassland and the contradiction between grass and livestock in this area, the state and Qinghai Province have invested a lot of manpower and material resources to plant artificial grassland in the source area of the Three Rivers, which has also curbed the further deterioration of the local ecological environment to some extent (*Chen et al., 2022*). In the process of grassland restoration, perennial grasses such as *Elymus sibiricus*, *E. nutans*, *Poa cryrnophila* and *Festuca sinensis* are often used to build artificial grassland. However, when perennial grasses are used to build artificial grassland, especially in the 5–8 years after planting, the grassland community develops to reverse succession (including soil reverse succession)
due to excessive grazing pressure (*She et al., 2023*). In order to solve this problem, many scholars have done a lot of research on grassland vegetation and soil restoration, which has provided rich theoretical guidance for the restoration of degraded ecosystems in the source region of the Three Rivers (*Xie et al., 2024*; *Chai et al., 2024*). Soil microorganisms have an important influence on plant health, soil productivity and ecosystem function, while they can increase the soil nutrients and regulate the soil enzymes as well (*Hu et al., 2016a*; *Hu et al., 2016b*; *Pan et al., 2018*), because they break down organic matter and release nutrients for plants to absorb (*Alori et al., 2024*). Especially in grassland covered with vegetation, soil microbial diversity plays a key role in maintaining soil health and quality (*Han et al., 2024*). Early studies have shown that, the establishment of artificial grassland can significantly increase the diversity and richness of plants and the content of soil organic carbon (*Zhao et al., 2016*; *Zhang et al., 2019a*; *Zhang et al., 2019b*; *Zhang et al., 2019c*; *Wu et al., 2010*), lower improve soil quality and fertility (*Feng et al., 2010*). It promotes the restoration of soil environment and microbial activity, and plays an important role in the restoration and reconstruction of degraded ecosystems. It is helpful to carry out effective and sustainable vegetation restoration in black-soil mountain area.

Phospholipid fatty acid is an important component of cell membrane of living microorganisms, which has the specificity of structure and species and genus, and can still exist after cell death. It is used to reflect the living biomass and flora structure of in-situ soil bacteria and fungi (*Lopes & Fernandes, 2020*). At present, the biochemical methods for studying soil microbial community structure mainly include plate counting method, Biolog substrate utilization analysis method and phospholipid fatty acid, PLFA) biomarker method, and the molecular biological techniques include nucleic acid hybridization technique and high-throughput sequencing technique (*Drenovsky et al., 2004*). The analysis of soil microbial community structure by phospholipid fatty acid method has the advantages of high accuracy, good stability and strong sensitivity. It is a rapid, reliable and reproducible analysis method, which can be used to characterize the dominant soil microbial community in quantity and is widely used in the study of microbial community structure (*Velasco et al., 2010*). Therefore, phospholipid fatty acids can be used to estimate the biomass of microbial living cells and qualitatively analyze the soil microbial community structure (*Wang et al., 2018a*; *Wang et al., 2018b*). Based on this, the present study monitored the vegetation characteristics, soil nutrients, and microbial community features of degraded black-soil mountain grasslands, artificial grasslands, and natural grasslands, and primarily addressed the following questions: (1) Compared with degraded black-soil mountain grasslands, what changes will occur in aboveground vegetation characteristics and soil nutrients after the establish of artificial grasslands? (2) How will soil microbial biomass and community structure change after the establish of artificial grasslands? (3) What are the associations between soil microbial communities and soil physicochemical properties in the three types of grasslands, and what are the key environmental factors influencing soil microbial communities?

## MATERIALS AND METHODS

### General situation of test site

This research was carried out in Zhiquegou (33 40′40″N, 99 42′59″E) in Wosai Township, Dari County, Qinghai Province, which is located at 4200 m and belongs to the continental plateau climate. The annual average temperature ranges from −0.1 °C to −3.5 °C, with the average temperature of −12.9 °C in January in Leng Yue, 9.1 °C in July in the hottest month, and the annual sunshine hours of 2,466.5 h.. The vegetation types in this area are rich and diverse, mainly alpine meadows. The main plants are Kobresia (*Carex parvula*), Kobresia (*Carex alatauensis*), Kobresia (*Carex capillifolia*), Androsace (*Pomatosace filicula Maxim.*), Potentilla (*Argentina anserina*), Polygonum (*Bistorta vivipara*), Cirsium (*Cirsium souliei* (Franch.) Mattf.and other species. The soil type is mainly alpine meadow soil (*Xie et al., 2024*).

### Experiment design

On June 10–20, 2019, in artificial grassland was planted in a typical black-soil mountain in Wosai Township, Dari County, Guoluo Tibetan Autonomous Prefecture. The selected grass species were *Elymus nutans* +*Poa crymophila* 'Qinghai'+*Festuca sinensis* 'Qinghai'. In August, 2023, the peak season of forage growth, the artificial grassland planted for 5 years and its surrounding degraded grassland and natural grassland in "black soil mountain" were selected as the research objects.The degraded grassland in "black-soil mountain" is in the stage of severe degradation, and the area of bare patches accounts for about 40%∼60% of the total grassland area. This experiment was carried out in the first half of August, 2023. In each of the above research objects (artificial grassland, "black-soil mountain" degraded grassland and natural grassland), three 1 m × 1 m quadrats were randomly set for vegetation investigation, and the interval between each quadrat should not be less than five m, totaling nine quadrats. The vegetation coverage, aboveground biomass and species richness in each quadrat were investigated and recorded respectively.

Vegetation coverage: the percentage of the total plot area occupied by the projected area of each plant. Above-ground biomass: plants were mowed in each quadrat level, and taken back to the laboratory to dry them to a constant weight at 65 °C, and calculate the yield. Species richness: the total number of species in the sample plot (*Zhao et al., 2023*). Soil samples were collected from each quadrat using a soil drilling sampler (with an inner diameter of 3.5 cm) to take five topsoil (0–10 cm) from each quadrat and mixing them evenly, and then combining them into one soil sample. Sieve was carried out with a two mm sieve to remove plant roots and impurities, and soil sample were divided into two parts. Some soil samples were air-dried due to the determination of soil nutrients. The other part of soil samples were taken back to the laboratory, labeled and stored in the ultra-low temperature refrigerator (−80 °C) for the determination of soil microbial community structure PLFA (*Zhou et al., 2024*).

## Data analysis
### Determination of soil physical and chemical parameters
Soil moisture content (SMC) and soil bulk density (BD) were determined by drying weighing method and ring knife method respectively. The pH value of soil (1:5) suspension was determined by Metter Toledo. Soil organic carbon (SOC) and total nitrogen (TN) were determined by C and N analyzers (Elementar, Langenselbold, Germany). The total phosphorus (TP) in soil was determined by alkali fusion method (*Zhao et al., 2024*).

### Determination of soil PLFA
In order to reveal the change characteristics of soil microbial phospholipid fatty acids after planting artificial grassland in black-soil mountain. According to the method of *Bossio & Scow (1998)*, the soil microorganism PLFA was determined. A total of 8 g of freeze-dried soil sample was accurately weighted, then placed in a 35 mL centrifuge tube; five mL of phosphate buffer, six mL of chloroform and 12 mL of methanol were added, shaken for 2 h, then centrifuged at 3,500 r/min, the supernatant was poured into a separatory funnel, and extraction and centrifugation were repeated. Then, chloroform and phosphate buffer were added, shaken for 2 min, and then allowed to stand overnight. The lower solution of the separatory funnel was collected, and then the solvent was evaporated under an oxygen-free $N_2$ flow. Using chloroform and acetone as eluents, the lipids and glycolipids were eluted by silica column (inner/outer tube diameter, CNWBOND Si SPE column, 500 mg, three mL). The polar lipids were eluted with methanol and collected in glass test tubes. The residue was shaken in a mixture of KOH and toluene: methanol to decompose basic methanol, and then incubated at 37 °C for 15 min. After cooling, it was neutralized with acetic acid, and fatty acid methyl ester (FAME) was extracted to hexane. The solvent was evaporated under oxygen-free N2 and then resuspended in hexane containing internal standard. The gas chromatograph (Agilent 7890B) and ESPRIT software system (MgIDI, Inc.) of Agilent 19091B column were used to identify and quantify the biomarkers of FAME. The concentration of each PLFA component was calculated based on the concentration of 19: 0 methyl internal standard (*Zhou et al., 2023*). PLFA can be used as a biomarker for changes in microbial biomass and community structure. According to the existing research results, PLFA biomarkers indicating specific microorganisms are shown in Table 1.

## Statistical analysis
The peak area internal standard curve method was used for quantitative determination of phospholipid fatty acids (PLFA). The content of PLFA is expressed as nmol $g^{-1}$ dry soil. Using Microsoft Excel 2010 to process data, SPSS 22.0 was used for the one-way ANOVA of soil physical and chemical properties and soil microbial PLFA content in degraded grassland, natural grassland and artificial grassland in "black-soil mountain", Duncan ($P = 0.05$), Pearson ($P = 0.05$) for correlation analysis and Canoco for Windows (5.0) for the principal component analysis (PCA) of soil physical and chemical properties and soil microbial PLFA content. Drawing was done with Origin 21.0 software.

**Table 1** Biomarkers of PLFA in artificial grassland of black-soil mountain.

| Microbial group | Phospholipids fatty acids signatures | Reference |
|---|---|---|
| Bacteria PLFAs (B PLFAs) | 14:0, i14:0, 15:0, i15:0, a15:0, 16:0, i16:0, 16:1ω7c, 16:1ω9c, 17:0, a17:0, i17:0, cy-17:0, i17:0 ω9c, 18:0, 18:1ω5c, 18:1ω7c, cy-19:0 | *Zhou et al. (2023)* |
| Fungi PLFAs (F PLFAs) | 18:2ω6c, 18:3ω6c, 18:1ω9c | *Sun et al. (2021)* |
| Arbuscular mycorrhizal fungi (AMF) | 16:1ω5c | *Dong et al. (2023)* |
| Actinomycetes (ACT) | 10Me-16:0, 10Me-17:0, 10Me-18:0 | *Bao et al. (2022)* |
| Gram-positive bacteria PLFAs (G+ PLFAs) | i14:0, i15:0, a15:0, i16:0, a17:0, i17:0 | *Francisco et al. (2022)* |
| Gram-negative bacteria PLFAs (G- PLFAs) | 16:1ω7c, 16:1ω9c, cy-17:0, i17:0 ω9c, 18:1ω5c, 18:1ω7c, cy19:0 | *Francisco et al. (2022)* |
| Cy:pre | (cy 17:0 + cy 19:0): (16:1 ω7c + 18:1ω7c) | *Fierer, Schimel & Holden (2003)* |
| Sat:mono | (14:0 + 15:0 + 16:0 + 17:0 + 18:0 + 20:0): (16:1ω7c+16:1ω5c+17:1ω8c+18:1ω5c+ 18:1ω7c+18:1ω9c + 18:2ω6c) | *Moore-Kucera & Dick (2008)* |

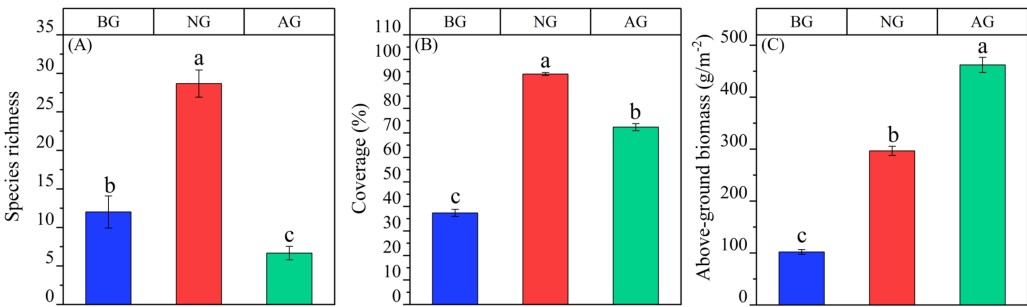

**Figure 1** **Quantitative characteristics of artificial grassland community in black-soil mountain.** AG, artificial grassland; BG, black-soil mountain degraded grassland; NG, natural alpine grassland. (A) Species richness; (B) coverage; (C) above-ground biomass. Values are the means $\pm$ SE ($n = 3$). Different lowercase letters indicate that there are significant differences among the three plots ($P < 0.05$).

# RESULTS

## Vegetation characteristics of artificial grassland in black-soil mountain

From different grassland types, the species diversity, coverage and aboveground biomass of the three grassland types were different ($P < 0.05$). The species richness of natural grassland is significantly higher than that of black-soil mountain degraded grassland and artificial grassland, and it is NG > BG > Ag (Fig. 1A). The coverage is NG > AG > BG (Fig. 1B); the aboveground biomass was AG > NG > BG (Fig. 1C).

## Physical and chemical properties of artificial grassland soil in black-soil mountain

The contents of soil organic carbon, total nitrogen and total phosphorus were significantly different in the three plots ($P < 0.05$) (Fig. 2). Specifically, the organic carbon content is NG

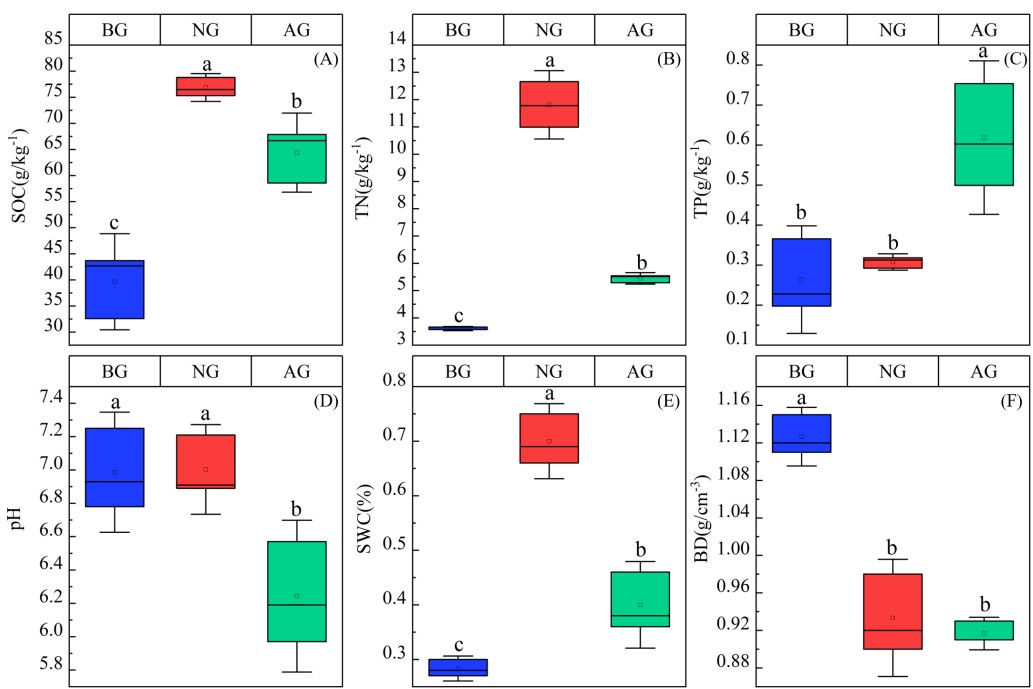

**Figure 2** **Physical and chemical properties of artificial grassland soil in black-soil mountain.** AG, artificial grassland; BG, black-soil mountain degraded grassland; NG, natural grassland. SOC, TN, TP, BD, Ph, and SWC represent the abbreviations of soil organic carbon, total phosphorus, bulk density, pH, and soil water content. Values are the means ± SE ($n = 3$). Different lowercase letters indicate that there are significant differences among the three plots ($P < 0.05$).

> AG > BG. The organic carbon and total nitrogen in artificial grassland are higher than those in black-soil mountain degraded grassland by 62.39% and 51.38% respectively, and lower than those in natural grassland by 22.47% and 116.69% respectively. However, the total phosphorus content in artificial grassland is 129.62% and 100% higher than that in black-soil mountain degraded grassland and natural grassland, respectively. establishment artificial grassland significantly reduced the pH value, which was NG >BG > AG, and there was no significant difference between black-soil mountain degraded grassland and natural grassland. Soil water content (SWC) is NG > AG > BG, while bulk density (BD) is BG > NG > AG, and there is no significant difference between NG and AG.

## Changes of PLFA species in artificial grassland in black soil mountain

There are 29 kinds of PLFA in the soil with the content more than 0.1 nmol g$^{-1}$ determined by PLFA method (Fig. 3). Twenty-six species were detected in black-soil mountain degraded grassland and natural grassland, while 27 species were detected in artificial grassland. Among them, microorganisms 17:1 iso w9c, 19:0 cyclo w9c and 16:0 10-methyl were not detected in artificial grassland and black-soil mountain degraded grassland. However, 19:0 cyclo w7c and 16:0 anteiso were not detected in natural grassland. Among them, the PLFA with the content more than 5% in artificial grassland and natural grassland are 16:1 w7c,
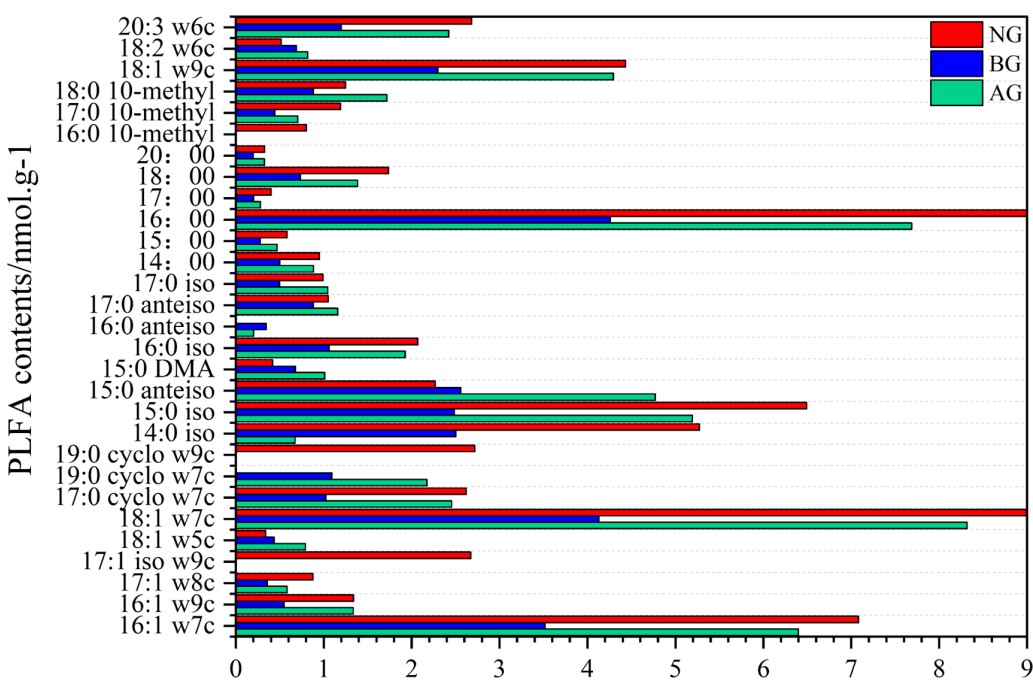

**Figure 3 Changes of PLFA species in artificial grassland in black-soil mountain.** AG, artificial grassland; BG, black-soil mountain degraded grassland; NG, natural grassland.

18:1 w7c, 14:0 iso, 15:0 iso and 16:00, which are G-PLFAs, G+ PLFAs and B PLFAs, which are the main groups in this plot. There is no PLFA with a content greater than 5% in the black-soil mountain degraded grassland.

There are differences in microbial community characteristics among the three plots (Fig. 4). The contents of total phosphatidic fatty acids, arbuscular mycorrhizal fungi (AMF), actinomycetes, B, F, G+ and G- in the artificial grassland were significantly higher than those in the black-soil Mountain degraded grassland (Figs. 3A, 3B, 3C, 3D, 3E, 3G, 3H), which were 74.34%, 102.5%, 84.09%, 73.15%, 70.9%, 44.46% and 98.5% respectively. The content of fungi in artificial grassland is significantly higher than that in natural grassland by 3.23%. However, the contents of other flora in artificial grassland were lower than those in natural grassland by 25.21% (total phospholipid fatty acids), 10.74% (AMF), 33.88% (actinomycetes), 28.45% (B), 16.64% (G+) and 33.04% (G-), respectively.

There were significant differences among the three plots in B:F,G+:G-,cy:pre and sat:mono ($P < 0.05$) (Fig. 5). B:F in artificial grassland tends to increase compared with black- soil mountain degraded grassland, but the difference is not significant, and it is significantly lower than that in natural grassland. However, G+:G- in artificial grassland is significantly lower than that in black-soil mountain degraded grassland, and tends to increase than that in natural grassland, but the difference is not significant. The ratio of cy:pre in artificial grassland was significantly higher than that in black-soil mountain degraded grassland and natural grassland, and there was no significant difference between black-soil mountain degraded grassland and natural alpine grassland. Sat:mono in artificial

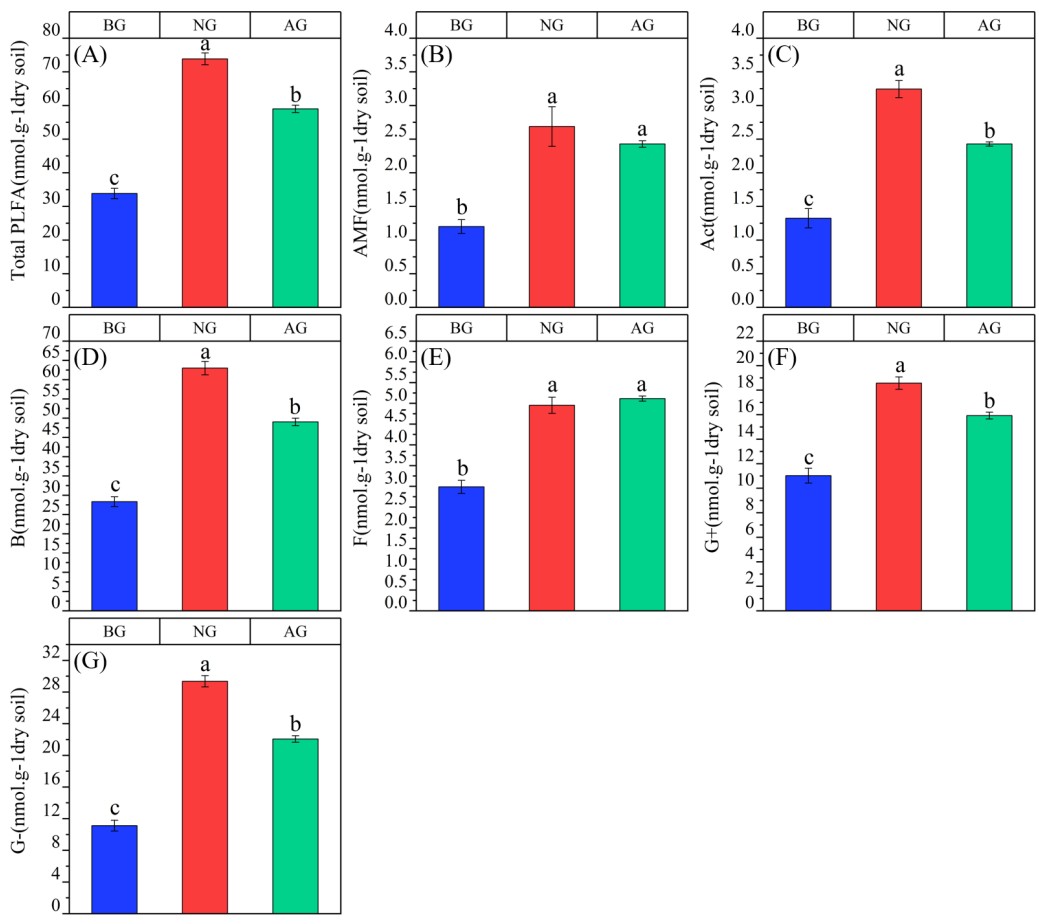

**Figure 4 Changes of the content and proportion of each flora in the artificial grassland in black-soil mountain.** AG, artificial grassland; BG, black-soil mountain degraded grassland; NG, natural grassland. (A) Total PLFA; (B) AMF, arbuscular mycorrhizal fungi; (C) Act, Actinomycete; (D) B, Bacterial; (E) F, Fungal; (F) G+, Gram-positive bacteria; (G) G-, Gram-negative bacteria. Values are the means ± SE ($n = 3$). Different lowercase letters indicate that there are significant differences among the three plots ($P < 0.05$).

grassland tends to decrease compared with natural grassland and black-soil mountain degraded grassland, and there is no significant difference between black-soil mountain degraded grassland and artificial grassland.

## Relationship between soil microbial biomass and community structure and soil physical and chemical properties

The soil microbial community structure of artificial grassland is influenced by vegetation characteristics and soil physical and chemical properties (Fig. 6). Total PLFA is positively correlated with vegetation coverage, organic carbon, total nitrogen and soil water content. Act, B, G- and G+ were positively correlated with vegetation coverage, organic carbon, total nitrogen and water content. AMF is positively correlated with vegetation coverage, aboveground biomass, organic carbon, total nitrogen and soil water content. Fungi is positively correlated with aboveground biomass and organic carbon, and negatively

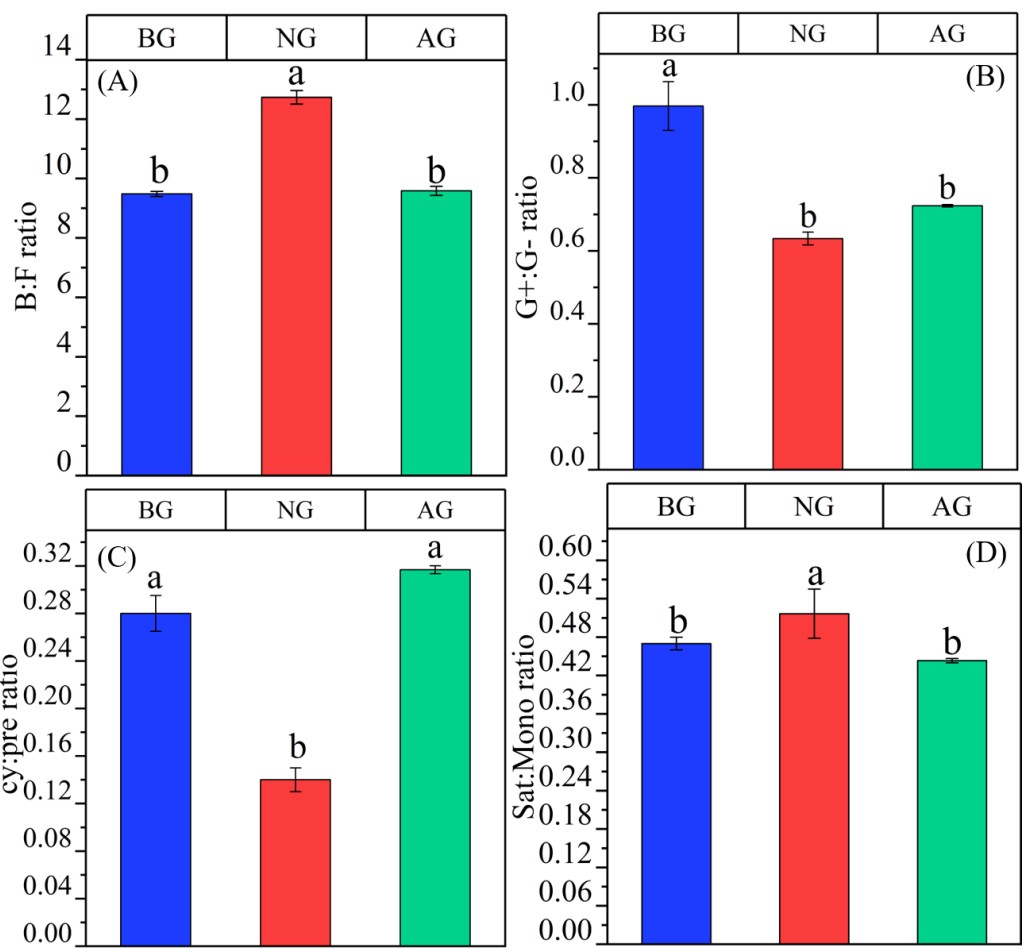

**Figure 5** **Stress index of artificial grassland in black-soil mountain.** AG, Artificial grassland; BG, black-soil mountain degraded grassland; NG, natural grassland. B:F ratio, ratio of bacterial to fungal PLFAs; G+, G− ratio: The ratio of gram-positive and gram-negative bacteria PLFA; cy:pre ratio, cy: 17: 0 + cy 19: 0 to 16: 1ω7c + 18: 1ω7c ratio; sat:mono ratio, Ratio of total saturated PLFA to total monounsaturated PLFA. Values are the means ± SE ($n = 3$). Different lowercase letters indicate that there are significant differences among the three plots ($P < 0.05$).

correlated with soil bulk density. Gram-positive bacteria/Gram-negative bacteria are negatively correlated with vegetation coverage, organic carbon, total nitrogen and soil water content. B:F is positively correlated with vegetation coverage, organic carbon and soil water content.

The RDA method was used to analyze the influencing factors of soil microbial community structure in the artificial grassland in black-soil mountain (Fig. 7). Physical and chemical properties such as pH, TP, SOC, SWC, BD and TN are taken as environmental parameters. The results of RDA show that the first axis and the second axis can explain 76.96% and 13.74% of the microbial community respectively, and all environmental variables can explain 90.7% of the microbial community variation, indicating that the first and second ranking axes well reflect the relationship between soil microorganisms and environmental

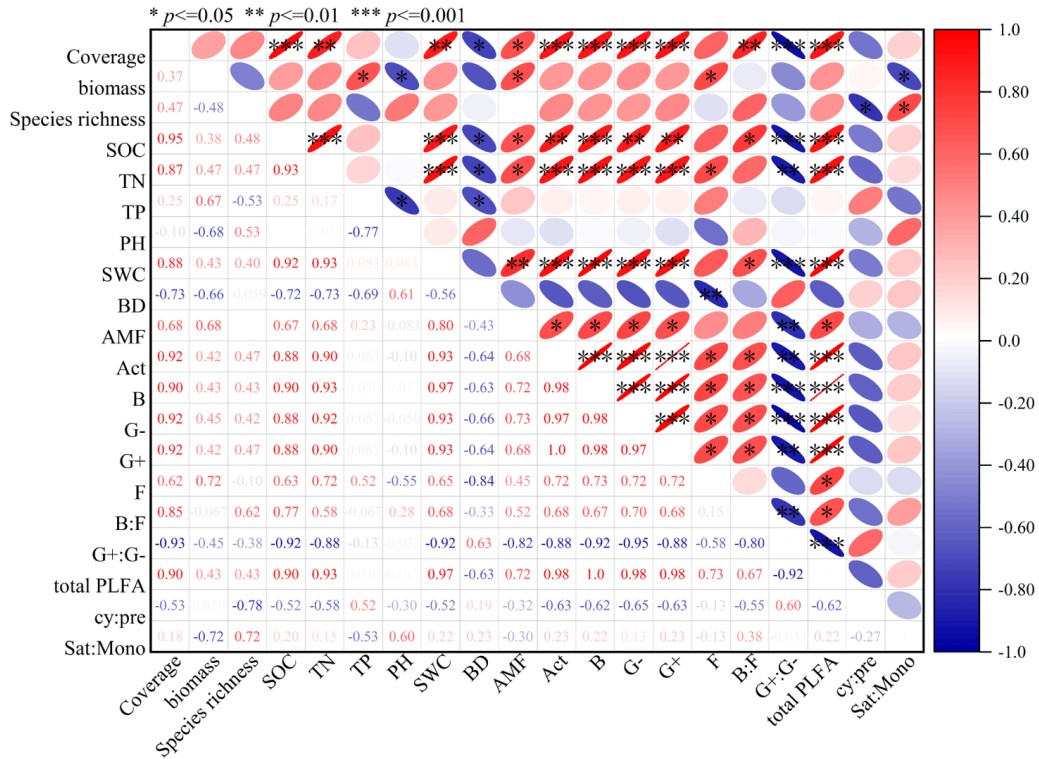

**Figure 6** Correlation of total PLFA, G+, G-, B, F, Act, AMF, F/B, G +/G-, cy:pre, sat:mono, soil physical and chemical characteristics and vegetation characteristics in artificial grassland in black-soil mountain. SOC, Organic carbon; TN, Total nitrogen; TP, Total phosphorus; pH, Soil pH; SWC, Soil water content; BD, Bulk density; AMF, arbuscular mycorrhizal fungi; Act, Actinomycete; B, Bacterial; G-, Gramnegative bacteria; G+, Gram-positive bacteria; F, Fungal; B:F ratio, ratio of bacterial to fungal PLFAs; G+:G− ratio, The ratio of gram-positive and gram-negative bacteria PLFA; cy:pre, Cy 17: 0+Cy 19: 0 to 16: 1 ω 7c+18: 1 ω 7c ratio; sat:mono, The ratio of total saturated PLFA to total monounsaturated PLFA.

factors, and this relationship is mainly determined by the first ranking axis. As shown in Fig. 7, the arrows of SOC and TN are the longest, which indicates that SOC and TN have the greatest influence on soil microbial biomass and community structure in these three plots, followed by BD, SWC, pH and TP.

The results of Monte Carlo test show that the important value (f) of soil physical and chemical properties decreases in the order of SOC > TN > BD > SWC > pH > TP (Table 2). In addition, the influence of SOC and TN is significant at the level of 0.001, and the explanation rates of these two soil physical and chemical properties are 71.6% and 14.6% respectively (Table 2). These results show that SOC and TN are the main driving factors affecting soil microbial biomass and community structure of artificial grassland in black-soil mountain.

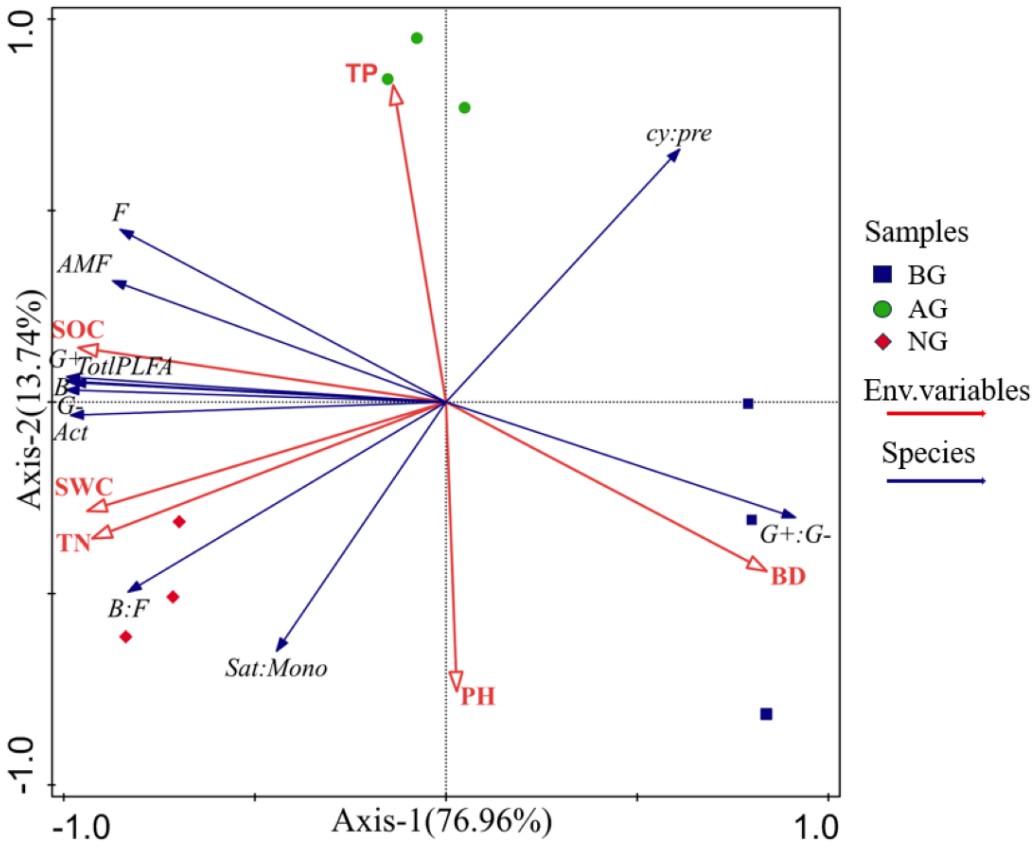

**Figure 7** Redundancy analysis (RDA) between soil microbial biomass and community structure and soil physical and chemical properties. Total PLFA, Total phospholipid fatty acid; Act, Actinomycete; B, Bacterial; F, Fungal; AMF, arbuscular mycorrhizal fungi; G+, Gram-positive bacteria; G-, Gram-negative bacteria; G+:G-, Gram positive-negative bacteria ratio; cy:pre; 17: 0 + cy 19: 0 to 16: 1ω 7 c + 18: 1ω 7 c ratio; Sat:Mono, the ratio of total saturation to total monounsaturated PLFA; pH, Soil pH; BD, Bulk density; SWC, Soil water content; TP Total phosphorus. TN, Total nitrogen.

**Table 2 Characteristics of influencing factors on horizontal changes of soil bacteria phylum.**

| Soil physicochemical properties | Importance ranking | Explains (%) | Contribution | Pseudo-F | P |
|---|---|---|---|---|---|
| SOC | 1 | 71.6 | 76.7 | 17.6 | 0.002 |
| TN | 2 | 14.6 | 15.7 | 6.4 | 0.004 |
| BD | 3 | 3.1 | 3.3 | 1.4 | 0.24 |
| SWC | 4 | 1.6 | 1.7 | 0.7 | 0.58 |
| pH | 5 | 0.9 | 0.9 | 0.3 | 0.796 |
| TP | 6 | 1.5 | 1.7 | 0.5 | 0.686 |

## DISCUSSION

Establishment artificial grassland is the most effective way to restore extremely black-soil mountain degraded grassland (*Li et al., 2014*). We found that compared with black-''-soil mountain degraded grassland, artificial grassland increased grassland coverage and

aboveground biomass, and decreased species richness (Figs. 1A, 1B, 1C), This is mainly because the artificial grassland is mainly Gramineae, and the change of aboveground biomass is mainly influenced by grasses, which account for a high proportion of grassland community biomass, in line with the "mass ratio hypothesis" (*DeMalach, Zaady & Kadmon, 2017*). In contrast, the change of plant species richness in grassland community depends on non-grasses to a great extent, which may be due to their high phylogenetic diversity and functional diversity (*Cadotte, Cardinale & Oakley, 2008*). This is consistent with the previous research results of *Deng et al. (2017)*. This not only changed the competition pattern of plants in the original grassland, but also accelerated the population renewal in the grassland plant community, causing changes in the community structure and function (*Dong et al., 2012*). Compared with black-soil mountain degraded grassland, establishment artificial grassland increased the contents of soil organic carbon, total nitrogen and total phosphorus (Figs. 2A, 2B, 2C), which is consistent with the research conclusion of *Wu et al. (2010)*. This may be attributed to the fact that the improvement of plant productivity will further increase the carbon input to the soil and reduce the carbon loss caused by erosion (*Hu et al., 2016a*; *Hu et al., 2016b*). The increase of total nitrogen and total phosphorus content may be caused by the increase of leguminous plant proportion and the decrease of erosion (*Van Der Heijden, Bardgett & Van Straalen, 2008*). The improvement of underground biomass of artificial grassland will increase soil microbial activity and the release of root exudates, thus improving the transformation of soil nutrients (*Wu et al., 2014*). Compared with natural grassland, the content of soil organic carbon and total nitrogen decreased after establishment artificial grassland, which is consistent with the research results of *Li et al. (2014)*. In Qinghai-Tibet Plateau, indicating that soil nutrients have not reached the level of natural grassland after establishment artificial grassland for 5 years. The study also showed that compared with black-soil mountain degraded grassland, BD decreased after planting artificial grassland (Fig. 2F), while the change of SMC was opposite to BD (Fig. 2E). Generally, increasing vegetation will improve the water-holding capacity of soil, thus increasing soil moisture (*Dong et al., 2012*). In addition, establishment artificial grassland significantly reduced the soil pH value (Fig. 2D), and acidic conditions were conducive to the decomposition of microorganisms, which was consistent with research (*Zhang et al., 2019a*; *Zhang et al., 2019b*; *Zhang et al., 2019c*).

Phospholipid fatty acid is a component of cell membrane of living organisms, and its community composition and diversity can reflect not only the biological activity of soil but also the ecological stress mechanism of soil (*Nüsslein & Tiedje, 1999*). Compared with other methods, phospholipid fatty acids can reflect the changes of microbial community more sensitively (*Zhang et al., 2019a*; *Zhang et al., 2019b*; *Zhang et al., 2019c*). In this study, 29 kinds of phospholipid fatty acid markers were detected from three plots (Fig. 3), which was different from the PLFA biomarkers detected by *Yan et al. (2020)*. who studied soil microbial diversity in the southwest of China. The reason for the difference may be that microorganisms are extremely sensitive to environmental changes, and it is also related to the selection of sampling points and determination methods. Secondly, the artificial grassland has a thick litter layer, which is the main source of energy for the survival of microorganisms. The thick litter layer increases the synergy between soil and

microorganisms, which may be the reason for the variety of PLFA biomarkers in artificial grassland (*Lange et al., 2015*). Compared with black-soil mountain degraded grassland, the content of microbial fatty acids in soil increased significantly after establishment artificial grassland (Fig. 4). This phenomenon can be attributed to the gradual establishment of vegetation and the continuous improvement of soil environment. These changes together create a more superior living environment for soil microorganisms, which is conducive to the production of microbial fatty acids, which is consistent with the research conclusion of *Angst et al. (2019)*. The study also found that bacteria are the main microbial groups in the artificial grassland in black-soil mountain (Fig. 3D). The metabolic activity of bacteria in soil is usually more vigorous than fungi, so their growth rate may be faster. Bacteria play an important role in decomposing organic matter and recycling nutrients (such as nitrogen, phosphorus and sulfur) (*Balasooriya et al., 2014*). Therefore, bacterial community plays a leading role in artificial grassland. Bacteria and fungi are two major groups in soil microbial community Fig. 5A (*Camenzind et al., 2021*). Studies have shown that the B/F ratio in soil not only reflects the degree of mineralization of carbon substrates in the soil, but also indicates that a higher value predicts better stability of the soil ecosystem (*Zhou et al., 2017*). In this study, the B/F ratio was highest in natural grasslands, followed by artificial grasslands, and lowest in degraded grasslands on black-soil mountain. This trend was consistent with the variation patterns of soil organic carbon and total nitrogen content. These results confirmed that the stability of the soil ecosystem in natural and artificial grasslands was higher than that in degraded grasslands on black-soil mountain, indicating that the soil quality improved significantly after the establishment of artificial grasslands. Gram-positive and Gram-negative bacteria can not only reflect the diversity characteristics of bacteria in the soil microbial community but also indicate the level of soil fertility (*Wang et al., 2018a*; *Wang et al., 2018b*). This study shows that the ratio of Gram-positive to Gram-negative bacteria in the degraded grassland of Heishan is higher than that in the artificial and natural grasslands, indicating that the soil microbial community in the degraded grassland of Heishan is under greater nutrient stress. Gram-positive bacteria can store carbon in their cells and react slowly to changes in carbon and nitrogen. Therefore, Gram-negative bacteria will decrease significantly when nutrient input is low (*Saetre & Bååth, 2000*). In this study, the soil nutrient content was the lowest in the black soil mountain, the number of Gram-negative bacteria decreased, and the nutrient stress was greater. In addition, the aboveground vegetation of the degraded grassland on the black soil mountain was relatively simple, and there were fewer litter, which could not provide rich nutrients for microorganisms. Therefore, microorganisms were subjected to greater nutrient limitations (*Agnihotri et al., 2023*). The G+/G- ratio can reflect the utilization of organic carbon in the soil, Among them, Gram-positive bacteria tend to consume newly imported carbon sources from plants first (*Byss et al., 2008*), While G-PLFAs tends to decompose those carbon sources that are difficult to decompose (*Huang et al., 2009*). After the artificial grassland was planted for five years, the proportion gradually decreased (Fig. 5B), which indicated that the availability of organic carbon in soil decreased with the recovery time, reflecting that the soil environment in artificial grassland was changing from "rich nutrition" to "poor nutrition". We also found that although the content of organic

carbon in artificial grassland is higher than that in degraded grassland, the availability of organic carbon is lower, which may be due to the increase of microbial content after planting artificial grassland, thus accelerating the decomposition of organic matter. This decomposition may lead to the decomposition rate of soil organic carbon exceeding the generation rate, thus reducing the effectiveness of soil carbon (*Wu et al., 2024*).

The restoration of grasslands depends on the interaction between plants, soil, and microorganisms. Plants stabilize the soil with their roots, promote the formation of organic matter, and improve the soil. At the same time, their root secretions also affect the activity of microorganisms (*Hernández-Cáceres et al., 2022*). Microorganisms, on the other hand, decompose plant residues and organic matter, providing nutrients that are available to plants and promoting their growth (*Li et al., 2022b*). Soil provides living environment for microorganisms, which affects their biomass and activities (*Tscherko et al., 2005*). Correlation and RDA results show that there is a significant correlation between soil microbial community, soil nutrients and plant community (Figs. 6 and 7 and Table 2). It was found that total phospholipid fatty acids, B, G- and G+ were positively correlated with vegetation coverage, organic carbon, total nitrogen and water content (Fig. 6), which was consistent with the research results of *Zhang et al. (2018)*. Vegetation coverage can represent the lush degree of grassland plants, and root exudates can promote the growth of soil microorganisms when plants grow and develop (*Kourtev, Ehrenfeld & Häggblom, 2002*). Higher plant diversity leads to higher soil microbial community diversity, which is due to the more diverse organic composition of litter and root exudates. Therefore, plant diversity and coverage can promote the growth of soil microorganisms (*Steinauer et al., 2015*). However, there are also different research results. *Tscherko et al. (2005)* think that plant species indirectly affect the microbial community by changing the soil environment, but not directly affect the soil microorganisms (*Tscherko et al., 2005*). Soil microbial community and soil physical and chemical properties are two important components of soil, and there is a close relationship and interaction between them. As the main source of energy required by microorganisms, soil organic carbon is considered as a key factor to regulate soil microbial biomass and community structure (*Grosso et al., 2018*). In addition, the growth activity of microorganisms is also limited by nitrogen. The higher the nitrogen content in the environment, the stronger the activity of microorganisms and the faster their reproduction (*Cao et al., 2010*). In this study, correlation analysis and RDA results show (Figs. 5 and 6) that the PLFA content of various soil microbial groups is positively correlated with soil organic carbon, total nitrogen and water content, in which organic carbon and total nitrogen are the most important factors affecting the PLFA content of soil microorganisms, and the explanation rate of organic carbon and total nitrogen is the highest (Table 2), which further confirms that the difference between soil organic carbon and total nitrogen is an important factor driving the change of soil microbial community biomass in artificial grassland in black mountain.

The diversity of PLFA biomarkers represents the diversity of microbial communities. Therefore, the composition and distribution characteristics of soil microbial community PLFA can be used to characterize soil quality. Some of the results obtained by this method cannot be obtained by other soil microbial diversity research methods. In view of the

inherent shortcomings of this method, the degraded grassland on the black-soil mountain will be monitored in real time in the future by combining high-throughput sequencing technology to obtain more comprehensive and complete information on the diversity of soil microbial communities in the degraded grassland on the black soil mountain, and to provide data support for the management of the degraded grassland on the black soil mountain.

## CONCLUSION

After establishment artificial grassland in black-soil mountain , some ecological functions such as aboveground biomass, vegetation coverage and soil nutrients were restored. The characteristics of microbial community structure in artificial grassland of "Black-Soil Mountain" were analyzed by phospholipid fatty acid technology (PLFA). Compared with black-soil mountain degraded grassland, establishment artificial grassland significantly increased the contents of total phospholipid fatty acids, actinomycetes, bacteria, fungi, gram-positive bacteria and gram-negative bacteria, and in addition, the ratio of G+/G– and sat:mono significantly decreased. SOC and total nitrogen are the main driving factors affecting the characteristics of soil microbial community. The study lays a strong foundation for future restoration monitoring frameworks and could inform adaptive grazing or reseeding strategies.

To sum up, the artificial grassland of "Black-Soil Mountain" has the value of popularization and application for restoring the vegetation of "Black-Soil Mountain", but the soil function of artificial grassland can not be restored to the state of natural grassland even after five years of restoration. After the establishment of artificial grassland, we should pay attention to its management and protection, and formulate a reasonable grassland utilization system to make its utilization sustainable. This study enhanced our understanding of grassland ecosystem function, and provided valuable insights for managing the restoration of degraded grassland in black-soil mountain.

### Funding

This work was supported by the Chief Scientist Program of Qinghai Province (2014-SF-101). The funders had no role in study design, data collection and analysis, decision to publish, or preparation of the manuscript.

### Grant Disclosures

The following grant information was disclosed by the authors:
Chief Scientist Program of Qinghai Province: 2014-SF-101.

### Competing Interests

The authors declare there are no competing interests.

## Author Contributions

- Lele Xie conceived and designed the experiments, performed the experiments, analyzed the data, prepared figures and/or tables, and approved the final draft.
- Yushou Ma conceived and designed the experiments, authored or reviewed drafts of the article, and approved the final draft.
- Yanlong Wang performed the experiments, prepared figures and/or tables, and approved the final draft.
- Yuan Ma performed the experiments, analyzed the data, prepared figures and/or tables, and approved the final draft.
- Xiaoli Wang conceived and designed the experiments, analyzed the data, prepared figures and/or tables, authored or reviewed drafts of the article, and approved the final draft.
- Ying Liu performed the experiments, prepared figures and/or tables, and approved the final draft.

## Data Availability

The original data is available in the Supplementary File.

## Supplemental Information

Supplemental information for this article can be found online at http://dx.doi.org/10.7717/peerj.19837#supplemental-information.

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
