# Peer review of "Characteristics of soil microbial phospholipid fatty acids in artificial, black-soil mountain degraded, and natural grasslands in the source region of the Three Rivers"

_PeerJ, doi:10.7717/peerj.19837_

## Round 0.1 · original submission · Major Revisions

· Academic Editor

Major Revisions

**Language Note:** The review process has identified that the English language must be improved. PeerJ can provide language editing services - please contact us at [email protected] for pricing (be sure to provide your manuscript number and title). Alternatively, you should make your own arrangements to improve the language quality and provide details in your response letter. – PeerJ Staff

Reviewer 1 ·

Basic reporting

pl see the general comments and minor comments.

Experimental design

pl see the general comments and minor comments.

Validity of the findings

pl see the general comments and minor comments.

Additional comments

This manuscript presents a meaningful and timely investigation into the restoration of degraded alpine grasslands in the Three Rivers Source Region through the establishment of artificial grasslands. By integrating vegetation characteristics, soil nutrients, and microbial community analyses using PLFA biomarkers, the authors provide a comprehensive understanding of ecosystem responses to restoration. The study is well-designed, the topic is of regional and global significance, and the findings offer practical insights for land management in sensitive alpine ecosystems. However, the manuscript still have some weakness need to be addressed before formal publication. So I would suggest accept it after considering the below comments.
1.Minor grammatical issues (e.g., “Establish artificial grassland” → “The establishment of artificial grassland”) should be revised throughout for improved readability and academic tone.
2. While the five-year restoration window is a good start, please consider discussing the need for longer-term monitoring to fully capture microbial community succession and functional recovery.
3.The authors might expand your discussion on how changes in microbial community structure relate to soil functions such as carbon sequestration, nutrient cycling, or plant–microbe interactions will be beneficial by the readership.
4. Could you please clarify whether the natural grassland used as a reference is undisturbed or has experienced some degree of anthropogenic influence. This affects the interpretation of recovery success.
5. In your introduction section, could you use some good reference for strength the robustness of the theory. DOI: 10.1016/j.jia.2024.11.008; DOI: 10.1016/j.agee.2022.108180; DOI: 10.3389/fpls.2022.947279; DOI: 10.1007/s11368-022-03209-9; DOI: 10.3390/rs13071279;

5. Please ensure consistent usage of terms such as “Black-soil Mountain,” “artificial grassland,” and “degraded grassland” throughout the manuscript.
6. The use of PLFA analysis is appropriate and well-executed. It might be beneficial to mention limitations of PLFA in taxonomic resolution and suggest potential future integration with DNA-based approaches for more detailed microbial community profiling.
7. The finding that microbial community ratios (e.g., G+/G-, B/F) are linked with soil nutrient status adds depth to the interpretation — this could be further explored in relation to ecosystem function and resilience.
8. The study lays a strong foundation for future restoration monitoring frameworks and could inform adaptive grazing or reseeding strategies. Maybe this section/ idea, could you please add into the discussion or conclusion section.

Reviewer 2 ·

Basic reporting

The manuscript “Characteristics of soil microbial phospholipid fatty acids in artificial grassland, black-soil mountain degraded grassland and natural grassland of the Three Rivers Source Region” present an interesting comparison of soil microbiome in three grasslands. The authors use a gradient of degradation, form a natural grassland to a degraded and a restored one, which is a god method to analyze the restoration potential and the differences in soil microorganism that are associated to each of the states.
To the current form of the manuscript, several improvements can be made:
General comments:
Use grassland within the entire manuscript. (e.g. line 44 “meadows” can be changed to areas). Please check the entire manuscript for the sentences to be correct (e.g. line 54 “degraded grassland (Ren et al., 2016), To alleviate” put a “.” after the ) ; line 72 “Pan et al., 2018).Because” put “,” after ) and lower because…; line 77 2010), Improve – lower improve - check for the entire manuscript to have the correct form of the sentences.
Check the entire manuscript for signs that need to be changed (e.g. line 110 Kobresia pygmaea, Kobresia.
Change “"Black soil beach" is a typical” into “This type is a typical …. Just to not repeat the same words.

Experimental design

Lines 93-98 – within these lines the authors present the aim of the research. The aim should be shorter and concise. Please reduce the length of the aim and remove the descriptive words. Try to present the aim as a comparison between three types of grassland in terms of vegetation, soil nutrients and microbial community characteristics. You can add a third research question (3) And what are the driving factors?. Make the word Establish as establish.
Line 113 Experimental design
Specify when the artificial grassland was established.
Line 120-134 – check the methodology style and the style of “mow” – line 126 into were mowed, “Collect” into Soil samples were collected.

Validity of the findings

Results section – there are a lot of results presented, and the significant differences are pointed out. The text for each figure explores completely the data. Remove the middle text form Figure 6, because it just make the graph harder to analyze and understand.
Both the Discussion and Conclusion sections are well organized. Discussion links the main trends with other international studies and present the benefits and changes due to restoration processes.

---

## Round 0.2 · accepted · Accept

· Academic Editor

Accept

I have reviewed the response letter and the revised manuscript and confirm that the authors have addressed all of the reviewers' comments. Both reviewers have checked the revised manuscript and are satisfied with the revisions provided by the authors.